# Chromosome-Level Assembly of Flowering Cherry (*Prunus campanulata*) Provides Insight into Anthocyanin Accumulation

**DOI:** 10.3390/genes14020389

**Published:** 2023-02-02

**Authors:** Dongyue Jiang, Xiangkong Li, Yingang Li, Shiliang Zhou, Qi Zhou, Xinhong Liu, Xin Shen

**Affiliations:** 1Institute of Tree Breeding, Zhejiang Academy of Forestry, 399 Liuhe Road, Hangzhou 310023, China; 2Novogene Bioinformatics Institute, Jiuxianqiao North Road, Beijing 100015, China; 3State Key Laboratory of Systematic & Evolutionary Botany, Institute of Botany, Chinese Academy of Sciences, Beijing 100093, China; 4Zhejiang Forestry Technology Popularization Station, 226 Kaixuan Road, Hangzhou 310019, China

**Keywords:** *Prunus campanulata*, genome assembly, comparative genomics, *PcMYB* gene family, anthocyanin accumulation

## Abstract

The flowering cherries (genus *Prunus*, subgenus *Cerasus*) are popular ornamental trees in China, Japan, Korea, and elsewhere. *Prunus campanulata* Maxim. is an important species of flowering cherry native to Southern China, which is also distributed in Taiwan, the Ryukyu Islands of Japan, and Vietnam. It produces bell-shaped flowers with colors ranging from bright pink to crimson during the Chinese Spring Festival from January to March each year. We selected the *P. campanulata* cultivar “Lianmeiren”, with only 0.54% of heterozygosity, as the focus of this study, and generated a high-quality chromosome-scale genome assembly of *P. campanulata* by combining Pacific Biosciences (PacBio) single-molecule sequencing, 10× Genomics sequencing, and high-throughput chromosome conformation capture (Hi-C) technology. We first assembled a 300.48 Mb genome assembly with a contig N50 length of 2.02 Mb. In total, 28,319 protein-coding genes were predicted from the genome, 95.8% of which were functionally annotated. Phylogenetic analyses indicated that *P. campanulata* diverged from a common ancestor of cherry approximately 15.1 million years ago. Comparative genomic analyses showed that the expanded gene families were significantly involved in ribosome biogenesis, diterpenoid biosynthesis, flavonoid biosynthesis, and circadian rhythm. Furthermore, we identified 171 *MYB* genes from the *P. campanulata* genome. Based on the RNA-seq of five organs at three flowering stages, expression analyses revealed that the majority of the *MYB* genes exhibited tissue-specific expression patterns, and some genes were identified as being associated with anthocyanin accumulation. This reference sequence is an important resource for further studies of floral morphology and phenology, and comparative genomics of the subgenera *Cerasus* and *Prunus*.

## 1. Introduction

Flowering cherries are popular ornamental trees in China, Japan, Korea, and elsewhere, belonging to the subgenus *Cerasus* of the genus *Prunus* in the Rosaceae family [1,2]. It is well known that Japan is the cradle of flowering cherry cultivars based on nine native species [3]; however, most of the wild flowering cherry plants are widely scattered over China, with 38 species and eight varieties in total [4]. In addition, the vast majority of these have not been fully exploited or developed.

*P. campanulata* Maxim. is an important species of flowering cherry native to southern China, which is also distributed in Taiwan, the Ryukyu Islands of Japan, and Vietnam. Compared with the white or light pink bulbs of cherry commonly seen in Japan, it produces bell-shaped flowers with colors ranging from bright pink to crimson [4]. *P. campanulata* is a typical early flowering species of flowering cherry, which often blooms during the Chinese Spring Festival from January to February each year, with a flowering period lasting up to 50 days [5]. It is the progenitor of many early flowering cherry cultivars in Japan, e.g., *P.* × *kanzakura* “Kawazu-zakura”, *P.* × *kanzakura* “Rubescens”, and *P.* × *introrsa* “Introrsa” [6,7]. In addition, *P. campanulata* has stable and robust resistance to cherry leaf spot [6], which is the most problematic fungal disease of the subgenus *Cerasus* by *Blumeriella jaapii* [8]. These extraordinary characteristics make *P. campanulata* suitable as an ornamental woody plant.

A high-quality genome sequence is essential for genome evolution and comparison [9]. Because of the gametophytic self-incompatibility of flowering *Prunus* species [10], flowering cherries generally have highly heterozygous genomes. To date, whole-genome sequences of four *Cerasus* species have been published, including *P. avium* [11], *P. yedoensis* [12], *Cerasus serrulata* [13], and *P. fruticosa* [14]. However, their genomes are complicated because of high heterozygosity or ploidy level, which has hindered their utility for genetic and breeding studies. In the current study, to obtain a reference-quality genome sequence, the *P. campanulata* cultivar “Lianmeiren”, with only 0.54% heterozygosity, was used; “Lianmeiren” is also an ornamental plant with magenta-colored double flowers (Figure 1). We generated a reference genome for *P. campanulata* using a combination of Pacific Biosciences single-molecule real-time (PacBio SMRT), 10× Genomics, and high-throughput chromosome conformation capture (Hi-C) technologies. Furthermore, we conducted a comparative analysis between *P. campanulata* and other *Prunus* species to understand the genome structure and evolution of *P. campanulata*.

## 2. Materials and Methods

### 2.1. Plant Sample

*P. campanulata* “Lianmeiren” individuals were cultivated in Shaowu City, Fujian Province, China (27°9’15″ N, 117°21’23″ E, altitude: 445 m). A plant with a height of 3 m and a diameter at breast height of 10 cm was used for sampling. Fresh young leaves, stems, and roots were collected in April 2017. Buds were collected in November 2017. Flowers were collected at the initial flowering stage, full flowering stage, and late flowering stage in 2018. All plant samples were stored at −80 °C.

### 2.2. Genome Size and Heterozygosity Analysis

*P. campanulata* genome size and heterozygosity were estimated based on *k*-mer analysis. A paired-end library with a short insertion size (350 bp) was constructed and used for sequencing on an Illumina HiSeq2500 platform. We estimated the genome size and heterozygosity of *P. campanulata* using approximately 21.8 Gb of the quality-filtered reads of Illumina data with Jellyfish v2.0 (k = 17) [15] based on the *k*-mer method, using the following formula: genome size = total *k*-mer number/average *k*-mer depth [16]. The distribution of distinct *k*-mers (k = 17) showed one sharp main peak and another very small peak at a depth of 53 and 26, respectively, suggesting low heterozygosity of the *P. campanulata* genome. The final analysis estimated the genome size of *P. campanulata* to be approximately 327.89 Mb, with approximately 0.54% and 48.67% heterozygosity and repeats, respectively (Appendix A). And the GC content was 40.18% (Appendix A).

### 2.3. Genome Sequencing

High-quality genomic DNA was isolated from the frozen leaves of *P. campanulata* using the DNAsecure Plant Kit (Tiangen, Beijing, China). A short-insert genomic library (300–350 bp) was prepared using the Illumina Library Preparation Kit (Illumina Inc., San Diego, CA, USA) according to the manufacturer’s instructions. The library was sequenced on the Illumina HiSeqXTen platform at Novo (Novogene Bioinformatics Institute, Tianjing, China). At least 10 μg of DNA was fragmented by Covaris g-TUBE^TM^ (Covaris Inc., Woburn, MA, USA) for a 20 kb insert size library construction. The single-molecule real-time bell (SMRTbell) sequencing library was prepared with the SMRTbell^®^ Express Template Prep kit (PacBio, Menlo Park, CA, USA), involving DNA concentration, damage repair, end repair, ligation of hairpin adapters, and template purification. The Sequel binding kit 3.0 was applied to bind sequencing polymerase to the SMRTbell library. Then, the single-molecular sequencing was performed on the Sequel (PacBio) sequencing platform using a Sequel Sequencing Kit v3.0 and SMRT Cell 1M v3 Tray (PacBio, Menlo Park, CA, USA). To build the 10× Genomics library, approximately 1 ng input DNA and the functionalized gel beads with unique barcoded primers were combined to form a “gel bead in emulsion” using a chromium automated microfluidic system (10× Genomics, San Francisco, CA, USA). The gel beads in the emulsion were amplified by PCR and then purified for Illumina sequencing on the Illumina HiSeqXTen platform. A Hi-C library was constructed using young leaves. First, young leaves were cross-linked with a 2% formaldehyde solution for 20 mins. Then, a restriction enzyme (*Hind* III) was applied to digest the samples at 37 °C. The DNA ends were marked with biotin-14-dCTP, and chromatin DNA was cyclized by ligation enzyme. DNA was purified by phenol-chloroform extraction. The library was sequenced on the Illumina HiSeqXTen platform.

### 2.4. Genome Assembly and Assessments

Genome assembly was conducted in a progressive manner. We first carried out a self-correction of errors of PacBio subreads using FALCON assembler v1.8.7 (parameters: length_cutoff_pr = 5000, max_diff = 120, max_cov = 130.) [17]. Then, the overlaps identified in all pairs of error-corrected reads were used to produce a directed string graph employing the “Myers” algorithm. Contigs were constructed based on the paths from the string graph. To eliminate the error rate of the preceding assembly, we used PacBio long-reads to polish assembled sequences using the consensus-calling algorithm Arrow v2.2.2 with default parameters [18], and then the corrected sequences were further polished using Pilon v1.20 [19], which mapped the Illumina reads to sequences with BWA v0.7.17 using default settings [20]. To order and orient these contigs into longer scaffolds, fragScaff [21] with default settings was used for 10× Genomics scaffolds extension as follows: (1) linked-reads were aligned to the consensus sequences of the PacBio assembly to form Super-Scaffolds using BOWTIE v2.2 with default parameters [22] and (2) qualitatively, the consensus sequences that were physically close to each other were supported by more linked-reads. By screening for linked-reads support, we filtered the consensus sequences without the linked-reads support and reserved the consensus sequences with the linked-reads support for the subsequent scaffolding. The Hi-C reads were mapped against the PacBio reads assembly using BWA v0.7.17 with the ‘-n 0′ option, which achieved mapping reads without any mismatches. Only read pairs aligned to the unique position of assembled sequences were preserved for Hi-C scaffolding. The strength of Hi-C interactions of contigs was quantified by standardizing the digestion sites of DpnII of read pairs on the genome sketch. LACHESIS v2e27abb [23] (parameters: CLUSTER N = 8, CLUSTER MIN RE SITES = 1157, CLUSTER MAX LINK DENSITY = 5, CLUSTER NONINFORMATIVE RATIO = 0) was used to cluster contigs into groups based on the strength of interactions, and then arranged and orientated in a linear order.

To verify the completeness of the assembled genome, the BUSCO database v3.0.2 [24] was used for the assessment, based on a benchmark of 1440 conserved plant genes. We also used CEGMA v2.5 to assess the assembly with 248 core eukaryotic genes [25].

### 2.5. Genome Annotation

Repetitive sequences, including tandem repeats and interspersed repeats, were identified in the genome assembly at both the DNA and protein levels. LTR_FINDER v1.06 [26], RepeatScout v1.05 [27], and RepeatModeler v1.0.11 [28] with default parameters were used to construct a de novo repeat sequences library. The de novo repeat sequences library was then mapped against Repbase [29] using Repeatmasker v4.0.7 withdefault parameters [30]. The mapped results from the above software were integrated based on the Uclust algorithm with the rule of 80-80-80 [31]. At the protein level, RepeatProteinMask v4.0.7 [32] with default parameters was used to identify the transposable element (TE) related proteins through mapping against the TE protein database from Repbase. Combing the results both at the DNA and protein level, the final TEs were obtained by removing redundant sequences.

The protein-coding genes were identified using de novo predictions-based, homology-based, and transcriptome-based approaches. For the de novo prediction, Augustus v2.5.5 [33], Genscan v3.1 [34], GlimmerHMM v3.0.1 [35], Geneid v1.4 [36], and SNAP [37] with default parameters were used to predict coding genes in the repeat-masked *P. campanulata* genome. For the homology-based prediction, the *P. campanulata* genome was mapped against the published sequences of *Arabidopsis thaliana* [38], *P. persica* [39], *P. mume* [40], *P. avium*, *Malus domestica* [41], *Pyrus bretschneideri* [42], and *Fragaria vesca* [43] using TblastN with an E-value cutoff of 1E-5 [44]. To improve the precision of spliced alignments, GeneWise v2.2.0 [45] with default parameters was used to filter all initially aligned coding sequences. For transcriptome-based prediction, a mixed library of five tissues (leaf, flower, bud, stem, and root) was constructed. The libraries were sequenced on an Illumina HiSeqXTen platform. A total of 32 Gb clean data were generated and de novo assembled with Trinity v2.11.0 using default settings (Grabherr et al., 2011), and then mapped against the genome assembly using TopHat v2.0.8 (parameters: -p 6 –max-intron-length 500,000-m^2^) [46]. Cufflinks v2.1.1 with default parameters [47] were used to assemble transcripts into gene models. The final consensus and non-redundant gene set were achieved using Evidence Modeler (EVM) v1.1.1 [48] with default parameters, which combined the genes predicted by the above three approaches. Moreover, PASA v2.4.1 under default settings [48] was used to correct the annotation results of EVM by generating untranslated regions. To annotate gene function, BlastP was used to blast the protein-coding gene sequences against Swiss-Prot (http://www.uniprot.org/ (accessed on 3 July 2019)) and TrEMBL [49] (E-value of 1 × 10^−5^). Protein domain annotation was performed using InterProScan v5.2 [50] and Hmmer v3.1 [51] with default parameters to search the InterPro (v66.0) and Pfam (v27.0) databases, respectively. Gene ontology (GO) terms [52] were assigned from the corresponding InterPro or Pfam results. The gene pathways were extracted by mapping genes against the Kyoto Encyclopedia of Genes and Genomes (KEGG) database [53].

The ribosomal RNA and micro-RNA genes in the *P. campanulata* genome were identified by searching the Rfam database (release 13.0) using BlastN (E-value of 1 × 10^−10^). The transfer RNA and small nuclear RNA were predicted by mapping against the Rfam database using tRNAscan-SE [54] and INFERNAL [55] with default settings, respectively.

### 2.6. Synteny Analysis

We applied McscanX v0.8 under default settings [56] to analyze the syntenic blocks and gene pairs between *P. campanulata* and *P. avium*, and *P. persica* genomes. MUMmer v3.23 software with default parameters [57] was used for detecting the genomic collinearity between *P. campanulata* and *P.* × *kanzakura* “Kawazu-zakura” genomes [58].

### 2.7. Gene Family Analyses and Evolution

To identify gene families in the *P. campanulata* genome, we used OrthoMCL v2.0.9 [59] to compare and cluster genes using all-to-all BlastP analysis (E-value of 1 × 10^−5^) among the protein sequences of 11 genomes, including *P. campanulata*, *P. persica*, *P. mume*, *P. avium*, *P. bretschneideri, M. domestica*, *F. vesca*, *Potentilla micrantha* [60], *Rosa chinensis* [61], *Rubus occidentalis* [62], and *A. thaliana*.

For detecting whole-genome duplication (WGD) events, first, collinear segments within *P. campanulata* and between *P. campanulata* and genomes of other related species (*P. avium*, *P. mume*, and *P. persica*) were identified using MCScanX v0.8 with default parameters [56]. Then, four-fold synonymous third-codon transversion (4DTv) values were calculated using the codeml tool in the PAML v4.8 with default parameters [63] according to the collinear segments. In addition, we calculated the number of synonymous substitutions per site (Ks) and non-synonymous substitutions per site (Ka) using protein-coding sequences. Then, the protein and nucleotide sequences were aligned using MUSCLE v3.8.31 with default settings [64], and Gblocks under default parameters [65] were used to exclude poorly aligned positions and divergent regions from the alignment. The branch-site likelihood ratio test was performed using codeml in the PAML v4.8 using default settings, and the ratio of Ka/Ks was calculated to identify positively selected genes (Ka/Ks > 1) in the *P. campanulata* genome. Based on the likelihood model implemented in CAFE v4.0 [66] with default parameters, expanded and contracted gene families were revealed in the *P. campanulata* genome.

### 2.8. Phylogenetic Analyses

The maximum likelihood (ML) analysis was performed using RAxML v8.2.12 [67], assuming the GTRCAT substitution model with 1000 bootstrap replicates based on sequence alignments using MUSCLE v3.8.31 with default parameters. Divergence times between the species were estimated using the Bayesian Markov Chain Monte Carlo algorithm implemented in PAML v4.8 with default parameters. In order to calibrate the phylogeny, fossil records of *A. thaliana* and *P. persica* from the TimeTree database (http://www.timetree.org (accessed on 10 September 2020)) were used for prior age calibration and secondary age calibration, respectively.

### 2.9. Identification of MYB Family Genes

All candidate protein sequences from the genome of *P. campanulata* were scanned by Hmmer v3.1 with default parameters employing the hidden Markov model profiles of the Myeloblastosis (MYB) domain (PF00249) downloaded from the Pfam (v30.0) databases. A database of transcription factors, constructed for the *MYB* gene family from *A. thaliana* downloaded from the Pfam (v30.0) database, was used as a query to search against the protein datasets of *P. campanulata* using BlastP. Subsequently, the protein datasets were analyzed with Swiss-Prot (http://www.uniprot.org/, accessed on 20 December 2020), using the method of auto-blasting two sequence sets to verify the domain composition. Based on the above methods, the preliminarily identified candidate sequences were determined by merging all the datasets and removing repeating sequences. The resulting candidate sequences were then confirmed by the Pfam (v30.0) and NCBI Conserved Domain Database (CDD, https://www.ncbi.nlm.nih.gov/cdd/ (accessed on 20 December 2020)). The sequences with MYB functional domains were retained for the following analysis.

### 2.10. Chromosome Distribution and Phylogenetic Analysis

The positions of *MYB* genes were validated by the General Feature Format (GFF) files of the *P. campanulata* genome. Additionally, the *MYB* genes from *P. campanulata* were located on the corresponding chromosomes and illustrated by the gene distribution and gene structure tools in TBtools software v1.09 [68]. MYB domain sequences from *A. thaliana* and *P. campanulata* were aligned with MAFFT v7.408 [69] under default parameters and adjusted manually with Se-Al v2.0 a11 under default parameters [70] if necessary. Phylogenetic trees were constructed in the MEGA X program [71] using the maximum likelihood (ML) method with 1000 non-parametric bootstrap replicates, LG substitution model, γ distributed (G) rates model among sites, and with other parameters under the default settings.

### 2.11. Transcriptomic Analysis

To analyze the expression levels of *MYB* genes, various tissues, including buds, mature leaves, stems, roots, and flowers at initial, full, and late flowering stages, were collected in three replicates to sequence using the Illumina HiseqXTen platform. The quality of sequencing data was controlled by Fastqc v0.11.9 with default parameters [72] and aligned to the reference genome from *P. campanulata* using Bowtie v2.2 with default parameters and TopHat v2.0.12 with parameters (‘mismatch = 2′). HTSeq v0.6.1 with default parameter [73] was used to count the reads numbers mapped to each gene. Then the values of fragments per kilobase of transcript sequence per million mapped reads (FPKM) of each gene were calculated based on the length of the gene and reads count mapped to the gene. Subsequently, differential expression analyses of various tissues and different flowering stages were performed using the DESeq R package v1.18.0 with default parameter [74]. Finally, heat maps of *MYB* gene expression were generated by using a plugin within TBtools software v1.09.

## 3. Results

### 3.1. Genome Sequencing and Assembly

*P. campanulata* (2n = 2x = 16) was used for whole-genome sequencing by a combination of Illumina short-read sequencing, PacBio SMRT sequencing, 10× Genomics sequencing, and Hi-C sequencing technologies. After adapter removal and low-quality reads filtering, a total of 53.47 Gb (163×) of clean reads from the HiSeqXTen platform, 42.73 Gb (130×) of subreads from the PacBio Sequel platform, 39.08 Gb (119×) of clean reads of 10× Genomics sequencing, and 42.20 Gb (130×) of clean Hi-C reads were generated (Appendix A).

By correcting the PacBio long reads, polishing assembled sequences, and extending 10× Genomics scaffolds, a genome assembly was generated, of which the total length was 300.48 Mb, comprising 492 scaffolds with a contig N50 size of 2.02 Mb, and the largest contig size was 11.75 Mb (Table 1). To anchor the scaffolds to chromosomes, LACHESIS software v2e27abb was performed to cluster 428 scaffolds into eight groups. As a result, the combined length of Hi-C contigs was 300.08 Mb, accounting for 99.87% of the total length of the assembled genome, and the length of the scaffold N50 was improved to 32.53 Mb, with the longest scaffold being 51.49 Mb (Appendix A). The final reference assembly comprised eight chromosome-scale pseudomolecules, corresponding to the haploid chromosome number of *P. campanulata* (Figure 2, Appendix A).

### 3.2. Assessment of Genome Quality

To evaluate the correctness of the *P. campanulata* genome assembly, we mapped short reads from the Illumina HiSeq sequencing data to the genome assembly using BWA v0.7.17. The results showed that the mapping rate was 97.41%, and the genome coverage was 99.50%. Furthermore, through single nucleotide polymorphism calling using Samtools v1.10 [75], the proportion of homologous single nucleotide polymorphisms was 0.0002%. To verify the completeness of the assembled genome, the BUSCO database v3.0.2 was used for the assessment, based on a benchmark of 1440 conserved plant genes. The analysis revealed that 96.6% of the conserved genes had complete gene coverage (including 92.8% single genes and 3.8% duplicated genes), 0.8% were fragmented, and only 2.6% were missing (Table 1). We also used CEGMA v2.5 to assess the *P. campanulata* genome with 248 core eukaryotic genes, of which 96.77% were detected, including 94.76% complete genes (Appendix A). In summary, the assessment of the correctness and completeness of the genome assembly indicated its high quality.

### 3.3. Genome Annotation

Repetitive sequences, including tandem repeats and interspersed repeats, were identified in the genome assembly at both the DNA and protein levels. Repetitive sequence length was 160,006,931 bp in total, accounting for 53.32% of the genome assembly. The most abundant repeat types were retrotransposons/class I elements, constituting 33.49% of the genome, of which long terminal repeat retrotransposons represented 31.15%. DNA transposons/class II elements and unknown repeat sequences represented 16.18% and 5.41%, respectively, of the whole genome (Figure 3 and Appendix A, Table 2).

We identified 28,319 protein-coding genes in the *P. campanulata* genome, with an average transcript length of 3135 bp, an average CDS length of 1226 bp, and an exon number of 4.75 per gene (Table 3). In addition, 26,352 protein-coding genes were annotated with ≥1 database, accounting for 93.1% of the annotated protein-coding genes (Appendix A). For the identification of non-coding RNA (ncRNA), a total of 2230 ncRNAs were predicted, including 562 microRNA, 721 tRNA, 330 rRNA, and 617 small nuclear RNA, accounting for approximately 0.10% of the *P. campanulata* genome (Appendix A).

### 3.4. Gene Family Analyses and Evolution

To understand orthologous relationships of genes between *P. campanulata* and ten other related species (*P. persica*, *P. mume*, *P. avium*, *P. bretschneideri*, *M. domestica*, *F. vesca*, *P. micrantha*, *R. chinensis*, *R. occidentalis*, and *A. thaliana*), after filtering gene sets, 286,011 genes were clustered into 27,921 orthologous groups (Figure 4B, Appendix A). By comparing the gene families among *P. campanulata* and three closely related *Prunus* species *(P. persica*, *P. mume*, and *P. avium*), there were 237 unique gene families specific to *P. campanulata*. The unique gene families were enriched in 37 GO categories and 12 KEGG pathways (*p*-value < 0.05) (Figure 4A). GO enrichment results mainly involved 3-methyl-2-oxobutanoate hydroxymethyltransferase activity, the tetrapyrrole biosynthetic process, and the single-organism cellular process (*q*-value < 0.05) (Figure 4C), and KEGG analyses revealed the gene families were enriched in ABC transporters, pantothenate and CoA biosynthesis, plant circadian rhythm, and flavonoid biosynthesis (*q*-value < 0.05) (Figure 4D). We also analyzed the collinearity between the *P. campanulata* and *P. avium*, and *P. campanulata* and *P.persica*. The results showed that there were a total of 172 and 120 syntenic blocks, which included 15,329 genes of *P. campanulata* and 15,014 genes of *P. avium* genomes, and 18,546 genes of *P. campanulata* and 18,000 genes of *P. persica* genomes, covering 54.18% and 35.51%, and 65.6% and 60.8% of their identified genes, respectively (Figure 3, Appendix A, Appendix A). To understand its contribution to early flowering cherry cultivars as a parent, we compared sequence synteny between the *P. campanulata* and *C. campanulata* haplotype, and the *P. campanulata* and *C. spachiana* haplotype, which were from *P.* × *kanzakura* “Kawazu-zakura” genome assembly [58]. A total of 227.4 Mb (76.01%) and 159.9 Mb (53.46%) of *P. campanulata* genomic sequences were syntenic with 223.5 Mb (85.24%) and 165 Mb (64.02%) of the *C. campanulata* haplotype (Appendix A) and the *C. spachiana* haplotype (Appendix A), respectively (Appendix A).

To clarify the phylogenetic position of *P. campanulate* among Rosaceae, 1090 single-copy orthologs of *P. campanulate* were used to construct a phylogenetic tree with nine species belonging to Rosaceae and *A. thaliana* as the outgroup. The ML tree revealed that *P. campanulate* formed a monophyletic clade with *P. avium* and was closely related to *P. mume* and *P. persica* (Figure 5A). Based on the phylogenetic relationship and fossil calibration, the node of our phylogenetic tree showed that *P. campanulate* and *P. avium* diverged approximately 15.1 million years ago. As representative species of *Prunus*, the clade with the four species diverged from the clade of *M. domestica* and *P. bretschneideri* approximately 64.1 million years ago. The divergent time estimation implied that *P. campanulate* was the most recently diverged species in *Prunus* (Figure 5A).

To determine the whole-genome duplication events in the *P. campanulata* genome, the 4DTv values among *P. campanulata*, *P. avium*, *P. mume*, and *P. persica* were estimated using their orthologous gene pairs. In the map, the peaks were grouped in 4DTv values of 0.02 and 0.55. One peak (4DTv approximately 0.02) indicated divergence events, including *P. campanulata* vs. *P. avium*, *P. campanulata* vs. *P. persica*, and *P. campanulata* vs. *P. mume*. Another peak (4DTv approximately 0.55) involving the four species suggested that a whole-genome or large-fragment duplication had occurred in their common ancestor (Figure 5B).

In order to comprehend the adaptive evolution of *P. campanulata*, we detected the genes under positive selection. Treating *P. campanulata* as the foreground branch, and *P. avium*, *P. persica*, and *P. mume* as background branches, 775 genes were identified as candidate genes under positive selection (*p*-value < 0.05) by an ML-based branch length test. GO and KEGG analyses showed that positive selection was especially detected in organic cyclic compound metabolism (*q*-value < 0.05) (Appendix A).

The protein-coding genes of 11 species genomes in the phylogenetic analysis were used for surveying expanded and contracted gene families during the evolution of *P. campanulata*. Based on the likelihood model implemented in CAFE with default parameters, 43 expanded gene families and 241 contracted gene families in total were revealed significantly (*p*-value < 0.05) in the *P. campanulata* genome, after divergence from *P. avium*. The GO and KEGG pathway analyses showed that the expanded gene families were significantly involved in ribosome biogenesis, diterpenoid biosynthesis, flavonoid biosynthesis, and circadian rhythm (*q*-value < 0.05) (Appendix A), and the contracted gene families were referred to pathogen/stress response (e.g., cyanoamino acid metabolism and plant–pathogen interaction), and phenylpropanoid biosynthesis (*q*-value < 0.05) (Appendix A).

### 3.5. Identification and Phylogenetic Analysis of MYB Family Genes

Based on genome-wide analysis of *MYB* genes in the *P. campanulata* genome and a BlastP search utilizing the Pfam database and Swiss-Prot database as queries, 171 PcMYB proteins with MYB DNA binding domains were identified. The resulting protein number is in the same range as those reported from 168 in *A. thaliana*, 163 in *P. persica* and 133 in *P. mume* in the Plant Transcription Factor Database (http://planttfdb.gao-lab.org/ (accessed on 12 January 2021)). The identified *PcMYB* genes were named *PcMYB1* ~ *PcMYB168* following the nomenclature of locations on the corresponding chromosomes (Appendix A). Notably, the proteins from four genes of the *PcMYB* gene family were closely related to MYB10 in *P. persica* [76] and *P. mume* [77], according to a phylogenetic tree constructed by using protein sequences, and they were named *PcMYB10a*, *PcMYB10b*, *PcMYB10c*, and *PcMYB10d*, respectively. The *PcMYB* genes were mapped to all eight chromosomes, but they were unevenly distributed. Chr1 contained the largest number of genes with 33 *PcMYB* genes, while Chr8 held the smallest number with 11 genes (Figure 6).

In order to explore the putative function of the predicted *P. campanulata MYB*s, the PcMYB proteins were assigned to the *A. thaliana* MYBs with known function and the most functional *MYB* gene family. The MYB proteins from *P. campanulata* (171 members) and *A. thaliana* (168 members) were used to construct a phylogenetic tree with a maximum likelihood method. According to the evolutionary relationship, the *MYB* genes’ functions appeared highly conserved across their clades, and closely related MYBs were recognized as the same or similar target genes, possessing cooperative, overlapping, or redundant functions. The results indicated MYB proteins were clustered into three separate subfamilies with branch support ranging from 17% to 100%. As shown in the unrooted tree, subfamily I included one AtMYB and six PcMYBs with branch support ranging from 74% to 100%; however, the function of AT5G04760.1 was not verified. As subfamily II only contained *P. campanulata* MYB members with branch support ranging from 73% to 100%, these MYB proteins may be unique to cherry. The majority of MYB members from *P. campanulata* and *A. thaliana* were distributed in subfamily III with branch support ranging from 17% to 100%, which contained many subgroups distributed loosely, as observed in other gene families generally (Figure 7).

### 3.6. MYB Gene Expression Profiles in Diverse Tissues

To analyze the expression levels of *MYB* genes in various tissues and at different flowering stages, including buds, mature leaves, stems, roots, and flowers at initial, full, and late flowering stages, the values of FPKM from unigenes were calculated and compared. The results of expression analyses, visualized in the form of heat maps, revealed that *P. campanulata MYB*s presented significant expression patterns in different organs and stages. The transcripts of 149 genes were detected in five tested tissues, while 22 genes were not exhibited. A total of 23 *PcMYB*s showed high levels of expression exclusively in buds, 20 in mature leaves, 14 in stems, 18 in roots, and 58 in flowers at the full flowering stage. The expression of 11 *MYB* genes was considerably higher in stems and roots, but lower in buds, leaves, and flowers. It was also found that five *MYB* genes were highly expressed in leaves, stems, and roots, but seldom transcribed in buds and flowers. Further investigation revealed that 19 genes uniquely exhibited high transcript abundance levels with expression at the initial flowering stage, 10 at the full, and 24 at the final flowering stage. Moreover, the expression levels of 31 genes increased gradually as the flowers matured, while those of 34 genes declined, and 20 genes first increased and then decreased (Figure 8, Appendix A).

## 4. Discussion

*Prunus* L. consists of over 200 species of deciduous and evergreen trees and shrubs, with many economically important species, such as cherries, peaches, plums, apricots, almonds, and others [78]. The plastid data support three main clades for *Prunus* L. *sensu lato*, including the subgenus *Prunus*, subgenus *Cerasus*, and subgenus *Padus*, which correspond to three groups based on inflorescence structure: the solitary-flower group, the corymbose inflorescence group, and the racemose inflorescence group, respectively [2,79,80]. As a transition type from the racemose group to the solitary group, subgenus *Cerasus* plants are valuable for genetic evolution and comparative genomics research of *Prunus*. Subgenus *Cerasus* involves 50–60 species distributed across the Northern Hemisphere and into the subtropics and tropics [2]. In the process of adaptation to diverse environments, these wild species have evolved various characteristics [81], which are paramount to cultivated cherry improvements in the future. However, the genome information of most of these wild *Cerasus* species is largely unavailable, and compared with the other two subgenera in *Prunus*, wild *Cerasus* species have a complex genome due to their variable ploidy levels and heterozygosity, which makes it difficult to obtain a high-quality reference genome [82]. The genome of *P. avium* (Satonishiki) was assembled using Illumina short-read technologies and the scaffold N50 length was only about 219.6 kb, which is incomplete and highly fragmented [11]. After that, the *P. yedoensis* genome was assembled based on Pacbio long-read and Illumina short-read technologies, but the contig N50 length was only 132 kb [12]. Yi et al. reported a high-quality reference genome of *C. serrulata* with a contig N50 length of around 1.56 Mb based on a combination of Nanopore, Illumina, and Hi-C sequencing technologies, which greatly improved the genome quality compared with the short-read-based draft genome of *P. avium* [13]. Recently, the genome of a tetraploid wild species, *P. fruticosa*, was assembled using Nanopore and the Hi-C sequencing platform with a contig N50 length of 533 kb [14] (Table 1). Nevertheless, because of the limitations of second-generation sequencing technologies and high-level ploidy and heterozygosity, these draft genomes are highly fragmented with a total size larger than expected, or fail to capture the true genome composition [83,84]. Here, we provide a high-quality genome of diploid cherry *P. campanulata* using SMRT Pacific Biosciences sequencing, Illumina sequencing, 10× Genomics sequencing, and Hi-C genome scaffolding, which can be used for genome evolution and comparative genomics research, and for identifying genes controlling important traits.

The genome size of *P. campanulata* is 300.08 Mb. It is 44.21 Mb and 66.42 Mb smaller than *P. avium* “Tieton” [85] and *P. fruticosa* [14], and 28.08 Mb and 34.68 Mb more than *P. avium* “Satonishiki” [11] and *C. serrulata* [13]. Besides the high-level heterozygosity and sequencing methods, as shown in Appendix A, TEs content is a leading reason for their genome size differences. Many studies have reported a strong correlation between genome size and TEs content [86,87,88]. *P. campanulata* is a species with magenta flowers, which is rare in the subgenus *Cerasus* [4]. In addition, it can flower very early due to a low chilling requirement [89]. These characteristics may have a major impact on its capacity for adaptation to the warm climate in subtropical and tropical regions. Our results indicate that the significantly expanded gene families in *P. campanulata* are mainly involved in ribosome biogenesis, diterpenoid biosynthesis, flavonoid biosynthesis, and circadian rhythm compared with other species (Appendix A). We also found that *P. campanulata*-specific genes were enriched in pathways participating in ABC transporters, protein export, pantothenate and CoA biosynthesis, phagosomes, glucosinolate biosynthesis, plant circadian rhythm, flavonoid biosynthesis, and biotin metabolism (Figure 4C, D). These gene families in *P. campanulata* may account for its phenotype and resistance to biotic/abiotic stress. *P. campanulata* is considered a progenitor of *P.* × *kanzakura*, which generated many early flowering cultivars with high ornamental value. Shirasawa et al. [58] reported the genome sequences of two early flowering cherry plants (*P.* × *kanzakura*), “Kawazu-zakura” and “Atami-zakura” for defining their origin, but due to missing the reference genome of *P. campanulata*, it is difficult to document their origin. Here, we compared the genome of *P. campanulata* with the *C. campanulata* haplotype and the *C. spachiana* haplotype from “Kawazu-zakura” genome assembly, respectively. The results show that the proportion of the P. campanulata and *C. campanulata* haplotype (85.24%) is significantly higher than the *C. spachiana* haplotype (64.02%), which indicated that *P. campanulata* is most likely a progenitor of *P.* × *kanzakura*. The MYB family is one of the largest transcription factor (TF) families in plants [90]. MYBs function in cell differentiation [91], hormone signal transduction [92], cell wall biosynthesis [93], and response to the biotic and abiotic stress of plants [94]. Furthermore, *MYB* genes play a crucial role in anthocyanin accumulation [95] and dormancy regulation [96], which are closely related to flower color and flowering time, respectively. Here, we identified a total of 171 *MYB* genes in the genome of *P. campanulata*. The number of *MYB* genes is similar to *P. persica*, *P. mume*, and *A. thaliana* in the Plant Transcription Factor Database. MYB10 is considered the master regulator in controlling anthocyanin accumulation in *Prunus* species, such as cherry (PavMYB10, PavMYB10.1) [97,98,99], peach (PpMYB10.1, PpMYB10.2, PpMYB10.3, PpMYB10.4) [76,100,101], apricot (PmMYBa1, PaMYB10) [77,102], and plum (PsMYB10) [103]. In this study, the homologous MYB10 genes in the *P. campanulata* genome exhibit tissue-specificity of expression. *PcMYB10a*, *PcMYB10c*, and *PcMYB10d* expression showed an obvious upward trend during flowering, whereas *PcMYB10b* expression was observed in the stem, root, leaf, and bud (Figure 7). In *Prunus*, such a situation also arises in the peach; PpMYB10.1, PpMYB10.2, and PpMYB10.3 are related to anthocyanin accumulation in fruit, whereas PpMYB10.4 and PpMYB10.2 are related to accumulation in leaves and flowers, respectively [100,104]. So *PcMYB10a*, *PcMYB10c*, and *PcMYB10d* may be involved in anthocyanin accumulation of flowers of *P. campanulata*, and *PcMYB10b* may be involved in anthocyanin accumulation of other tissues. Additionally, *PcMYB10a* and *PcMYB10c* performed high expression at the initial flowering stage, whereas *PcMYB10d* showed high expression at the full flowering stage, which means that they may contribute to flower coloration at different stages. These results will facilitate elucidating the regulation of anthocyanin accumulation in the flowering cherry.

## Figures and Tables

**Figure 1 genes-14-00389-f001:**
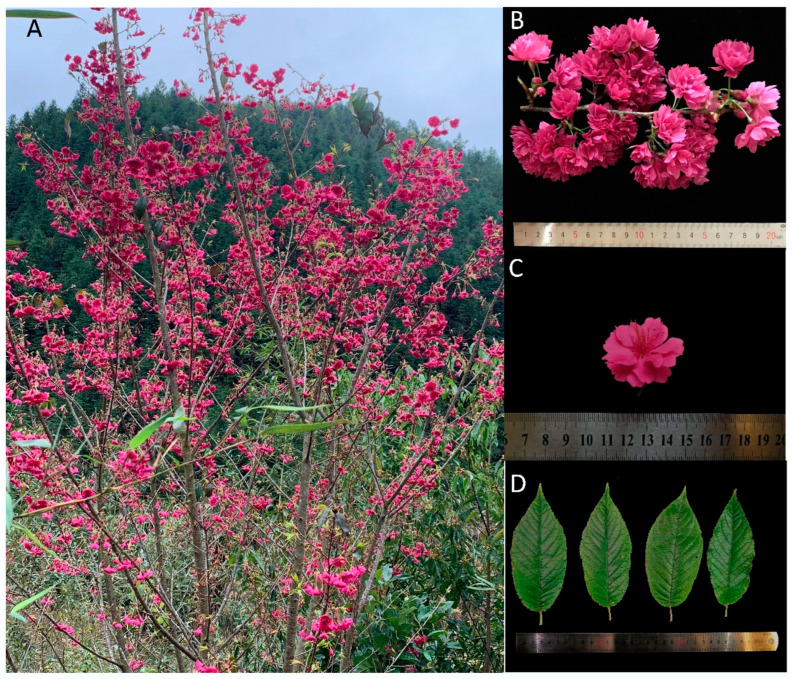
Morphology of *P. campanulata* cultivar “Lianmeiren”. (**A**) *P. campanulata* tree grows in Shaowu City, Fujian Province. (**B**) Flower branch and inflorescence. (**C**) Flower. (**D**) Leaves. The scale unit of the ruler is a centimeter.

**Figure 2 genes-14-00389-f002:**
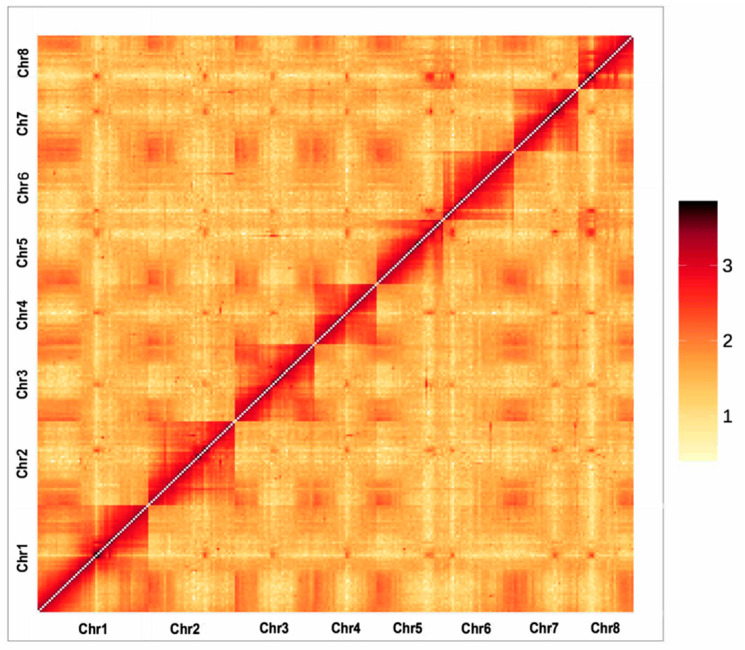
Hi-C contact map revealing extensive hierarchical chromatin interactions in the genome of *P. campanulata*.

**Figure 3 genes-14-00389-f003:**
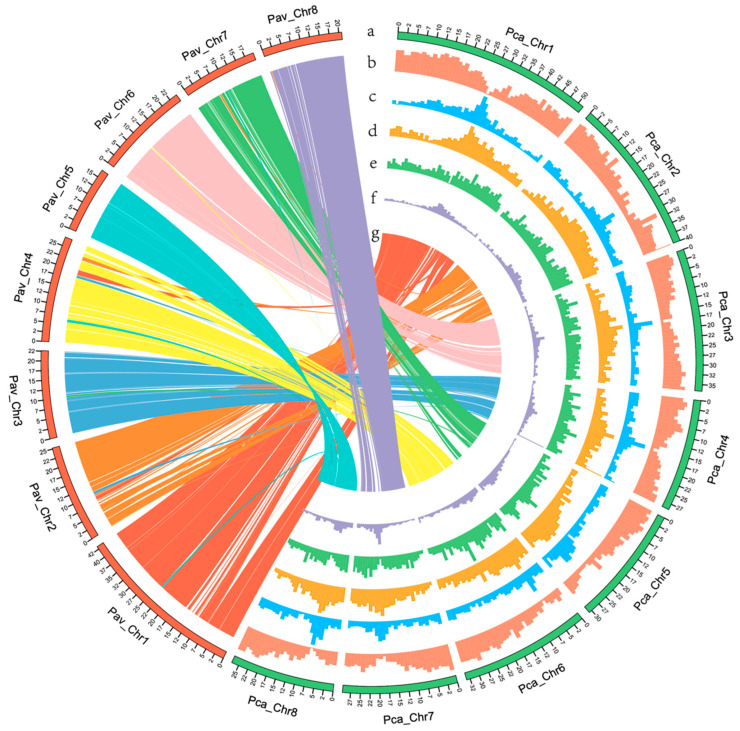
Circular diagram depicting the characteristics of *P. campanulata* genome. (**a**) Genome of *P. campanulata*. (**b**) Gene density in 1 Mb sliding windows. (**c**) GC content in 1 Mb sliding windows. (**d**) Repeat density in 1 Mb sliding windows. (**e**) Copia density in 1 Mb sliding windows. (**f**) Gypsy density in 1 Mb sliding windows. (**g**) Syntenic relationships of gene pairs between *P. campanulata* and *P. avium* genomes using the best-hit method. Pca: *P. campanulata*; Pav: *P. avium*.

**Figure 4 genes-14-00389-f004:**
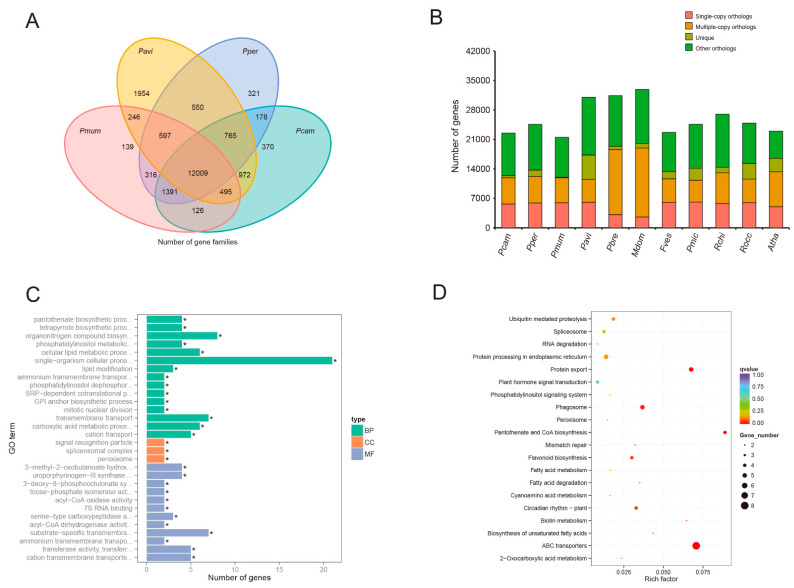
Gene family analysis of the genome of *P. campanulata*. (**A**) The unique and shared gene families among four species. Pcam: *P. campanulata*; Pper: *P. persica*; Pmum: *P. mume*; Pavi: *P. avium*. (**B**) Comparison of orthologs between *P. campanulata* and other angiosperm species. Pcam: *P. campanulata*; Pper: *P. persica*; Pmum: *P. mume*; Pavi: *P. avium*; Pbre: *P. bretschneideri*; Mdom: *M. domestica*; Fves: *F. vesca*; Pmic: *P. micrantha*; Rchi: *R. chinensis*; Rocc: *R. occidentalis*; Atha: *A. thaliana*. (**C**) GO enrichment analysis of *P. campanulata*-specific genes. (**D**) KEGG enrichment analysis of *P. campanulata*-specific genes.

**Figure 5 genes-14-00389-f005:**
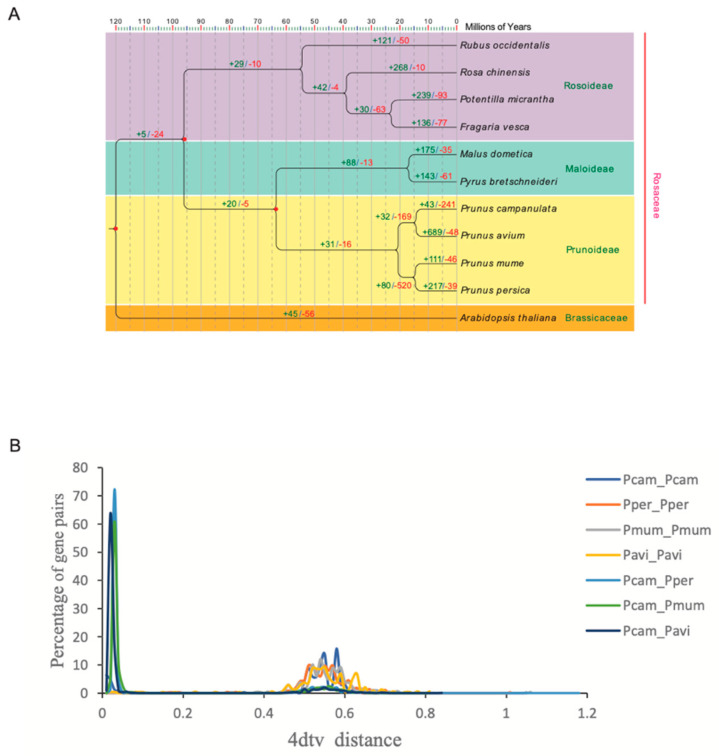
Evolution of the genome and gene families of *P. campanulata*. (**A**) Phylogenetic tree with the number of gene families displaying expansion and contraction from 11 species for estimating divergence time. The numbers on the branches show the expanded (green) and contracted (red) gene family numbers among all gene families. (**B**) Distribution of 4DTv values in *Prunus* species. Pcam: *P. campanulata*; Pper: *P. persica*; Pmum: *P. mume*; Pavi: *P. avium*.

**Figure 6 genes-14-00389-f006:**
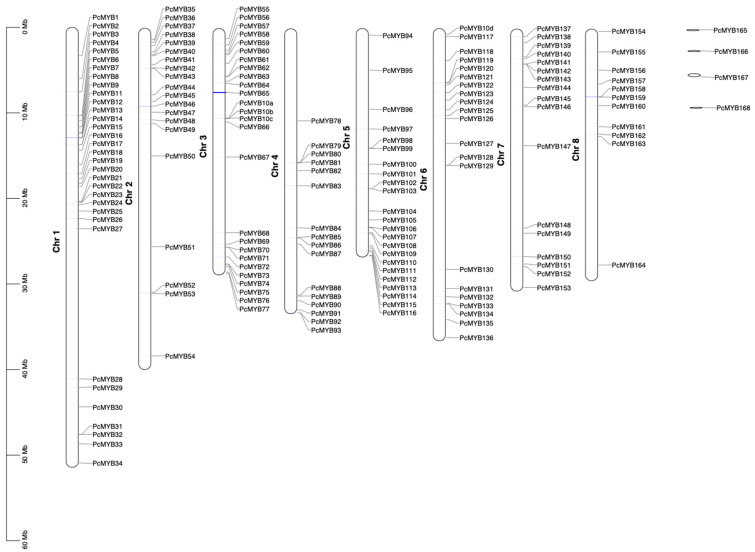
Chromosomal localizations of the *MYB* genes. The bars represent the *P. campanulata* chromosomes. The scale bar on the left indicates the physical distance of the *MYB* genes on the chromosomes. Four *MYB* genes on the right are located on the scattered scaffolds.

**Figure 7 genes-14-00389-f007:**
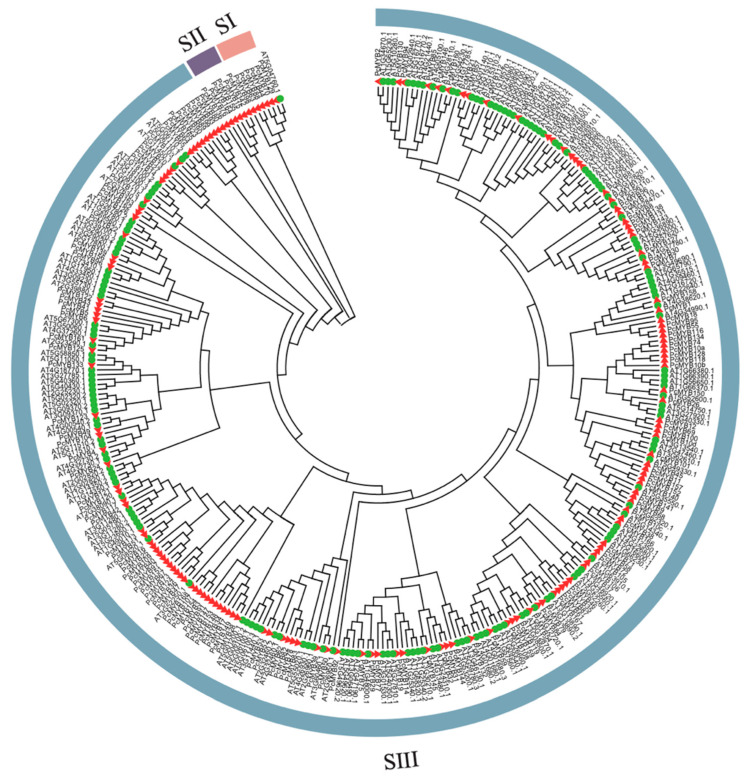
Phylogenetic relationships of the MYB proteins from *P. campanulata* and *A. thaliana*. The red triangle denotes the protein from *P. campanulata*, and the green circle denotes the protein from *A. thaliana*.

**Figure 8 genes-14-00389-f008:**
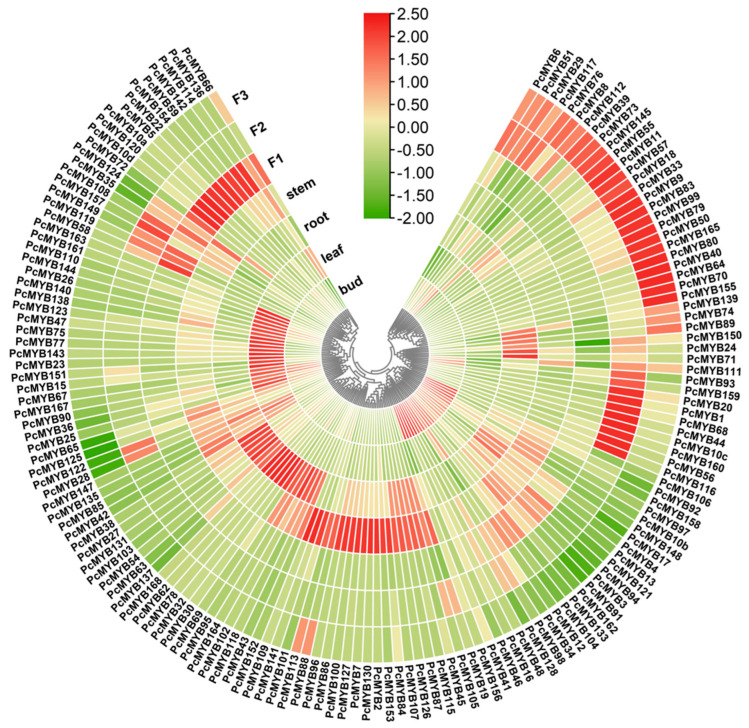
Expression profiles of the *MYB* genes in various tissues. F1: flowers at the initial flowering stage; F2: flowers at the full flowering stage; F3: flowers at the late flowering stage. The color bar in the middle indicates log2 expression values, with red representing high levels and green representing low levels of expression.

**Table 1 genes-14-00389-t001:** Comparison of the genome assembly among the four cherry species.

Parameter	*P. campanulata*	*P. avium* “Satonishiki”	*P. yedoensis*	*C. serrulata*	*P. fruticosa*
Sequencing platform	PacBio, Illumina, 10× Genomics, Hi-C	Illumina	PacBio, Illumina	Nanopore, Illumina, Hi-C	Nanopore, Hi-C
Heterozygosity percentage	0.54%	—	—	1.67%	—
N50 length (contigs) (Mb)	2.02	—	0.133	1.56	0.533
N50 length (scaffolds) (Mb)	33.44	0.219	0.198	31.12	43.82
No. of protein-coding genes	28,319	43,349	41,294	29,094	58,880
Chromosome level/Total size of assigned scaffolds (Mb)	Yes/283.62	Yes/191.70	No	Yes/252.25	Yes/—
No. of scaffolds	370	10, 148	3, 185	67	—
Total length of assembled scaffolds (Mb)	300.48	272.36	323.78	265.4	—
No. of contigs	687	51,877	4292	182	1275
Total size of assembled contigs (Mb)	299.15	—	318.74	263.16	366.5
Busco assessment	C:96.6% [S:92.8%,D:3.8%], F:0.8%,M:2.6%,n:1440	C:96% [S:78.3%,D:17.7%], F:1.8%,M:2.2%,n:956	N/A	C:94.7% [S:83.8%,D:10.9%], F:1.86%,M:3.47%,n:1614	C:96.4% [S:94.1%,D:2.3%], F:1.3%,M:2.3%,n:1614

**Table 2 genes-14-00389-t002:** Summary of repetitive sequences in the genome of *P. campanulata*.

Type	De novo + Repbase ^1^		TE Proteins ^2^		Combined TEs ^3^	
	Length(bp)	% in Genome	Length(bp)	% in Genome	Length(bp)	% in Genome
DNA	46,577,999	15.52	8,655,257	2.88	48,545,281	16.18
LINE ^4^	5,778,324	1.93	2,431,202	0.81	7,025,267	2.34
SINE ^5^	14,544	0	0	0	14,544	0
LTR ^6^	92,381,320	30.79	22,553,446	7.52	93,460,219	31.15
Unknown ^7^	16,239,753	5.41	0	0	16,239,753	5.41
Total	157,524,477	52.49	33,566,604	11.19	160,006,931	53.32

^1^ De novo + Repbase: annotated de novo by RepeatModeler, RepeatScout, and LTR_FINDER; ^2^ TE proteins: annotated by RepeatProteinMasker based on RepBase protein database; ^3^ Combined TEs: merged results from approaches above by removing overlaps; ^4^ LINE: long interspersed nuclear element; ^5^ SINE: short interspersed nuclear element; ^6^ LTR: long terminal repeat; ^7^ Unknown: repeat sequences cannot classify.

**Table 3 genes-14-00389-t003:** Summary of protein-coding genes annotation in *P. campanulata* via three methods.

Method	Gene Set	Number	Average Transcript Length (bp)	Average CDS Length (bp)	Average Exons Per Gene	Average Exon Length (bp)	Average Intron Length (bp)
De novo	Augustus	21,837	2714.22	1201.26	4.98	241.3	380.31
	GlimmerHMM	35,772	6382.00	796.8	3.41	234.01	2322.32
	SNAP	22,256	3128.31	721.53	4.17	172.89	758.43
	Geneid	31,597	3987.08	914.41	4.49	203.78	881.1
	Genscan	20,493	8917.35	1370.01	6.42	213.25	1391.37
Homolog	*Arabidopsis_thaliana*	26,286	2299.76	1045.17	3.79	276.13	450.47
	*Fragaria_vesca*	21,218	3082.18	1347.82	4.46	302.29	501.45
	*Malus_domestica*	25,135	2556.85	1263.62	4.2	300.84	404.10
	*Prunus_avium*	27,359	2019.39	891.10	3.68	242.34	421.46
	*Prunus_mume*	24,960	2764.26	1273	4.5	283.05	426.38
	*Prunus_persica*	29,626	2176.78	1082.37	4.01	269.93	363.61
	*Pyrus_bretschneideri*	21,837	2893.41	1292.81	4.42	292.21	467.42
RNAseq	PASA	70,332	2899.39	1019.99	4.79	212.94	495.89
	Cufflinks	47,625	5826.71	2195.12	6.17	355.93	702.80
EVM		30,644	3031.10	1161.66	4.53	256.69	530.25
Pasa-update		30,431	3022.99	1181.12	4.56	258.99	517.31
Final set		28,291	3135.45	1226.50	4.75	258.41	509.54

## Data Availability

The data presented in this study are available at NCBI BioProject under accession no. PRJNA884816.

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
