# Peer review of "Chromosome-Level Assembly of Flowering Cherry (Prunus campanulata) Provides Insight into Anthocyanin Accumulation"

_genes, 2023, doi:10.3390/genes14020389_

Round 1
Reviewer 1 Report (New Reviewer)
Jiang et al. present a Comprehensive genomic study of Prunus campanulata using a wide range of bioinformatics approaches and gene expression of the entire MYB family of P campanulata.
The authors have made an effort to sequence and analyze the entire genome at the level of chromosomal individualization in P. campanulata, which is unpublished information in the scientific literature. Additionally, the authors also analyzed the expression of all 171 MYB genes.
The story is interesting, and it should be considered by Genes.
1 1. The authors should inform the statistical support of the phylogenetic tree in figure 7.
2 2. I just felt that the authors could have discussed a bit more about the expression patterns of MYBs genes. Mainly regarding PcMYB10a, PcMYB10b, PcMYB10c and PcMYB10d.
Author Response
Point 1: The authors should inform the statistical support of the phylogenetic tree in figure 7.
Response 1: Thanks for your suggestion. I have inform the statistical support of the phylogenetic tree in the description of results.
Point 2: I just felt that the authors could have discussed a bit more about the expression patterns of MYBs genes. Mainly regarding PcMYB10a, PcMYB10b, PcMYB10c and PcMYB10d.
Response 2: Thanks. According to your suggestion, I enhanced discussion about the expression patterns of MYBs genes in the discussion part.
Reviewer 2 Report (New Reviewer)
Kindly do the minor corrections as suggested in the attached file.
Moderate revision in English language is required.
Improve the quality of Fugue 4, 5 and 6

Author Response
Thanks for your review and suggestions. I have revised the manuscript according to your suggestion in the attached file. And I replaced the Figure2, 3, 4, 5, 6, 7, 8 with high quality pictures.
This manuscript is a resubmission of an earlier submission. The following is a list of the peer review reports and author responses from that submission.
Round 1
Reviewer 1 Report
The manuscript “Chromosome-Level Assembly of Flowering Cherry (Prunus 2 campanulata) Provides Insight into Anthocyanin Accumulation” has merits but there are changes required in the manuscript. At places, manuscript is not clear about what they did, also additional comparisons required to make the MS meaningful.
1. The chemistry is not clear. What chemistry was used at what point? Which version of RSII/sequel etc. Authors need to provide a table for all the relevant methods used for generating sequence, along with vital statistics like depth of data, number of reads, size of library etc,\.
2. The choice of comparison does not make sense. Why only P. avium was chosen for synteny analysis? Other Prunus species have already been sequenced and are of better contiguity as done using third generation sequencers compared to the P. avium which is done using Illumina.
3. Authors themselves are saying that Kawazu-zakura is a possible progenitor. “Kawazu-zakura” has already been sequenced (10.1093/dnares/dsab026). So, comparative analysis using this genome would have provided better insights about the progenitor of Prunus campanulata.
4. Description of the bioinformatic process is very fleeting. Please provide details. Authors need to mention tools used at each step. If the tool is run at default parameter, please mention it, if the parameters were changed, please mention that also.
5. K-mer analysis for genome size estimation is not described fully. Authors need to mention the tool used for identifying the k-mers (k=17) (Like Jellyfish, KAT) as only GCE (Genomic character estimator) is mentioned which take k-mer file as an input to estimate the genome size.
6. L107-108: The description is insufficient. How much of data was >6kb? What was the data amount after correction using daligner? Why say upto 99.99%? This means that at places accuracy was lower, if it is so please provide details.
7. L266-267: Since the authors used illumine dataset for polishing, the SNPs are not expected! Please explain why they are mentioning it.
8. L151-156: Handling of transcriptome is not clear to me. It seems like authors have aligned transcriptome data to the genome assembly and then used cufflink to assemble it. Since the genome is not complete, many of the genes will not be present in assembled genome. These genes might be present in transcriptome, but with the approach used by authors, these genes will also be lost.
9. L461-470: Considering that authors have generated a lot of data I was expecting maximum number of genes, but I am surprised to see low number of genes as compared to earlier assemblies. This has to do with the poor assemblies also as given in table 1. Here only comparable assemblies are of P. fruticosa and C. serrulata while rest of the two are highly fragmented assemblies (hence the point number two: see above). The data about C. serrulata is not complete, please complete it. Authors need to investigate why number of genes are ~58000 here? This section is not written well. They need to compare genes predicted with these two species and see the reason for change in number of genes. They cannot call their number of genes low as a natural reason (See line 462).
Reviewer 2 Report
The authors submitted a well written paper on the genome assembly of Prunus campanulata. Their strategy allowed them to obtain a good draft genome, and to detect gene expansions and reductions.
However, the authors compared the genome structure and evolution of P. campanulata, and then focused on the the expression of MYB genes, but do not offer other insights into this plant species. So I think that this paper is similar to other studies, as it thorougly describes a novel genome, but I fail to see an hypothesis driving this paper, or scientifically interesting conclusions that may be drawn from this study (apart from stating the expression levels of specific MYB genes).
I believe that this paper would have a better fit as a genome report, rather that as a research article.
Author Response
Thanks for your comments and suggestions. I added a comparison of genomes between P. campanulata and P. × kanzakura “Kawazu-zakura” for exploring whether P. campanulata is a progenitor of P. × kanzakura in the revision. If you are interested in it, please review the manuscript, I really appreciate your suggestions.
Round 2
Reviewer 1 Report
The manuscript “Chromosome-Level Assembly of Flowering Cherry (Prunus campanulata) Provides Insight into Anthocyanin Accumulation” has been revised as per expectations.
1. The chemistry is still not clear. Sequel can be HiFi or RSII. Seems like authors are not sure about what they did, which raises doubts about data generated and its presentation.
2. The comparison is still not there. The circos diagram (Fig 3) is still not updated. It is not clear what actually they did to compare, which raises question about the integrity of data and its analysis.
3. Q3 has been answered.
4. The efforts have not changed clarity yet. Pipeline would have helped. At places the description is confusing and does not reflect what actually has been done and why? For example, L171: The authors have used denovo assembler followed by a reference based assembly which is very confusing again raising the doubts about their understanding of analysis.
5. Q5 answered.
6. Q6 has been smudged and the answer is not satisfactory at all. It is not clear what they did and why they did it. What was the accuracy? What was the data left?
7. Point number 7 have not been addressed. I cannot understand how SNP can be used for looking at accuracy of assembly? Please provide the basis of assumption and the pipeline which recommends it.
8. Point 8 has not been addressed and smudged again. Section 2.11 is a mess and does not tell you what they did? They have used TopHat and Bowtie, then what happened to Trinity?
9. The answer is not satisfactory for Number 9 also. If the data is not available, they need to take raw files and run it to find how many genes they are getting for this data set vs their dataset. Number of genes reported are in general low, so it should have been addressed.
I was hoping that authors will understand the pitfalls and overhaul the manuscript in a constructive manner. They have go beyond reviewers comment and make manuscript intelligible, but they did not do much to inspire confidence.
Reviewer 2 Report
The authors used a thorough approach to assembly and annotate a low heterozygous genome of P. campalunata, and also included a transcriptomic sequencing process to improve gene prediction.
I only have a question: how did the authors obtained the final set numbers (Table 3?)
Author Response
Response: Thanks. First, we used de novo predictions-based, homology-based, and transcriptome-based approaches to identify protein-coding genes. Then, the gene sets based on the above three methods were combined and reduced redundancy using EVidenceModeler. To obtain the final set, PASA was used to correct the annotation results of EVidenceModeler according to the information of untranslated regions and variable splicing based on the transcriptome assembly.